# Targeting Latent HIV Reservoirs: Effectiveness of Combination Therapy with HDAC and PARP Inhibitors

**DOI:** 10.3390/v17030400

**Published:** 2025-03-12

**Authors:** Hasset Tibebe, Dacia Marquez, Aidan McGraw, Sophia Gagliardi, Cailyn Sullivan, Grace Hillmer, Kedhar Narayan, Coco Izumi, Adleigh Keating, Taisuke Izumi

**Affiliations:** 1Department of Biology, College of Arts & Sciences, American University, Washington, DC 20016, USA; ht8146a@american.edu (H.T.); dm6732a@american.edu (D.M.); amcgraw@american.edu (A.M.); sg1978a@american.edu (S.G.); cs0973a@american.edu (C.S.); gh2297a@american.edu (G.H.); kn0119a@american.edu (K.N.); cizumi@american.edu (C.I.); akeating@american.edu (A.K.); 2District of Columbia Center for AIDS Research, Washington, DC 20052, USA

**Keywords:** latency, Kick and Kill, HDAC inhibitors, PARP inhibitors, NK cells

## Abstract

The “Kick and Kill” strategy, which aims to reactivate latent HIV reservoirs and facilitate the clearance of reactivated HIV-infected cells, has yet to achieve a functional cure due to the limited efficacy of current latency reversal agents. This study evaluates the combination efficacy of histone deacetylase (HDAC) inhibitor with poly(ADP-ribose) polymerase (PARP) inhibitor in latency reversal and immune-mediated clearance. Latently infected J-Lat cells and dual-fluorescent HIV-infected primary CD4 T cells were treated with the HDAC inhibitor (vorinostat) and one of four PARP inhibitors (olaparib, rucaparib, niraparib, or talazoparib). PARP inhibitors, when administered alone, showed no latency reversal activity. However, when combined with vorinostat, their efficacy increased threefold compared to vorinostat alone. This effect was mediated by the inhibition of tankyrase, a PARP superfamily member, which modulates the Hippo signaling pathway. In HIV_GR670_-infected primary cells, the combination reduced the reservoir size by 67%. In addition, talazoparib alone significantly reduced actively infected cells by 50%. Talazoparib-treated peripheral blood mononuclear cells co-cultured with K562 cells demonstrated enhanced NK-cell-mediated cytotoxicity, with a 10% reduction in K562 cell viability. These findings demonstrate that combining HDAC and PARP inhibitors augments latency reversal and reservoir reduction. With both the HDAC inhibitors and PARP inhibitors used in this study approved by the FDA for cancer treatment, this combination therapy holds strong potential for rapid clinical integration, contingent upon the confirmation of efficacy and safety in ongoing in vivo studies.

## 1. Introduction

As of 2024, there have been seven cases where individuals have achieved remission from human immunodeficiency virus (HIV), referred to as being cured, by undergoing stem cell transplantation, particularly from donors with the CCR5-∆32 mutation [1,2,3]. Despite these successes, a universally applicable method for HIV eradication or functional cure has not yet been developed, often due to co-existing conditions such as leukemia. Long-acting antivirals and a new class of antivirals targeting virion maturation [4,5], effective against heavily multi-drug-resistant mutants, have been developed recently [6]. While these treatments improve the quality of life for people living with HIV, further research is needed to explore more accessible cure strategies. Research on HIV functional cure has rapidly progressed with the development of new strategies [7,8]. Among the several promising strategies for HIV eradication, the Kick and Kill therapy has been proposed as a notable approach [9]. This approach is contingent upon the reactivation of latently infected cells (“Kick”) in the presence of combined antiretroviral therapy (cART) through latency reversal agents (LRAs) and the subsequent elimination of forcibly reactivated HIV-infected cells by immune systems or another intervention (“Kill”). Several pathways related to cis- and trans-regulating factors have been targeted, and latently infected cells have been successfully reactivated on an individual in vitro basis [7]. This strategy emphasizes the increasing interest in pairing LRAs with broadly neutralizing antibodies (bnAbs) for improved virus suppression [10,11] and combining LRAs, including histone deacetylase inhibitors, with potent bnAbs that show promise in diminishing the size and functionality of HIV reservoirs [12,13]. Histone deacetylase (HDAC) inhibitor has been extensively researched in in vitro models and has led to clinical trials for reactivating HIV latent cells, which has shown a moderate reduction in the reservoir size by the combined approach with immunovaccination [14].

A recent article published in the Journal of Virus Eradication summarized the clinical potency of LRAs on the HIV-1 reservoir, screening more than 5000 publications [15]. This report indicates that reactivation after LRA treatment occurred in 78% of studies, mostly within 24 h of treatment initiation with HDAC inhibitors. A small decrease in reservoir size was observed in three studies using a combination of immune checkpoint inhibitors and HDAC inhibitors. However, combination LRA strategies have been infrequently studied, and no synergistic reactivation was observed. In addition, the negative influence of selected HDAC inhibitors on immune cells, including natural killer (NK) cells, which are crucial effector cell populations contributing to the reactivated reservoir clearance, has recently been reported [16,17,18,19,20]. Given that the sole use of HDAC inhibitors may not effectively reactivate most latent reservoir cells, and that an immune response is crucial for the clearance of reactivated cells, the enhancement of latency reversal efficacy by HDAC inhibitors to reduce the reservoir size in vivo is necessary.

In this study, we aimed to explore a novel target: the inhibition of tankyrase, a member of the 17 poly(ADP-Ribose) polymerase (PARP) family proteins. Tankyrases (TNKS1 and TNKS2) are two homologous proteins that belong to the PARP family of proteins [21,22]. These proteins regulate a variety of cellular processes, including (1) Hippo signaling and (2) Wnt/β-catering signaling [23,24] (Figure 1). The Hippo signaling pathway regulates cell proliferation and apoptosis by controlling the nuclear localization of YAP, which translocates into the nucleus and binds to TEAD transcription factors to activate gene expression [25,26,27]. In the Wnt/β-catenin signaling pathway, stabilized β-catenin translocates into the nucleus and forms a transcriptional complex with TCF/LEF transcription factors, driving the expression of target genes involved in cell fate determination and proliferation [28,29]. We demonstrated that FDA-approved PARP inhibitors enhanced the latency reversal efficacy of HDAC inhibitors by approximately three-fold in J-Lat cells. PARP inhibitor also reduced the latently infected cell populations in in vitro models of HIV-latently infected primary cells, despite those inhibitors themselves lacking reactivation efficacy in latently infected cells [30]. In addition, PARP inhibitors enhance NK-cell cytotoxic activity, which may improve the clearance of reactivated HIV-1 infected cells.

Tankyrase controls several downstream signaling pathways, including β-catenin signaling and Hippo signaling. Tankyrase suppresses the β-catenin destruction complex by modifying a key component of the complex called Axin through polyADP-ribosylation. This modification leads to Axin degradation, thereby disrupting the complex’s ability to target β-catenin for destruction and allowing for increased β-catenin signaling. In addition, tankyrase promotes the degradation of angiomotin (AMOT) proteins through polyADP-ribosylation, marking AMOT for ubiquitin-mediated proteasomal degradation. AMOT proteins are known to bind and sequester YAP, the primary effector of the Hippo pathway, in the cytoplasm, preventing its nuclear translocation and transcriptional activity. By targeting AMOT for degradation, tankyrase effectively releases YAP, enabling its accumulation in the nucleus and the subsequent activation of Hippo-responsive gene expression.

## 2. Materials and Methods

### 2.1. Plasmid DNA Construction

The plasmid DNA, pHIV_GR670_, used to visualize actively and latently infected cell populations in primary CD4 T cells, was constructed based on the pHIV_GKO_ backbone (Addgene, Watertown, MA, USA) [31]. The EF1α-derived miRFP670nano3-encoding gene [32,33], flanked by XhoI restriction enzyme sites at both ends, was synthesized by Twist Bioscience and integrated into the XhoI-digested pHIV_GKO_ plasmid backbone. The plasmid DNA is extracted and purified by NucleoBond Xtra Maxi Plus EF kit (TaKaRa, Shiga, Japan).

### 2.2. Cell Culture

HEK293T cells, kindly provided by Dr. Zachary Klase at Drexel University, were cultured in Dulbecco’s Modified Eagle’s Medium (DMEM) (Cytiva, Marlborough, MA, USA) supplemented with 10% fetal bovine serum (FBS) (Gibco, Waltham, MA, USA), 1% penicillin-streptomycin-glutamine (Gibco, Waltham, MA, USA), and 1% GlutaMax (Gibco, Waltham, MA, USA) (D10) in 10 cm Cell Culture Dish (CELLTREAT, Ayer, MA, USA) at manufacture recommended appropriate seeding density at 37 °C in a 5% CO_2_ environment. J-Lat cell lines (J-Lat 6.3, 8.4, 9.3 and 10.6), provided by BEI Resources at the American Type Culture Collection (ATCC, Manassas, VA, USA), were cultured in RPMI 1640 medium (Cytiva, Marlborough, MA, USA) with the same supplements as D10 medium (R10) in T25-flask (CELLTREAT, Ayer, MA, USA) at manufacture recommended appropriate seeding density at 37 °C in a 5% CO_2_ environment. Human peripheral blood mononuclear cells (PBMCs) were isolated from the blood of a healthy donor using density gradient separation. The cells were then cultured in complete medium (R10) supplemented with 1 × MEM Non-Essential Amino Acids (Gibco, Waltham, MA, USA) and 1 × Sodium Pyruvate (Gibco, Waltham, MA, USA) in T25-flask (CELLTREAT, Ayer, MA, USA) at manufacture recommended appropriate seeding density at 37 °C in a 5% CO_2_ environment. Before culturing, recombinant human IL-2 protein (R&D Systems, Minneapolis, MN, USA) was added at a final concentration of 40 U/mL. K562 cells, kindly provided by Dr. Marta Catalfamo at Georgetown University, were cultured in R10 medium in T25-flask (CELLTREAT, Ayer, MA, USA) at manufacture recommended appropriate seeding density at 37 °C in a 5% CO_2_ environment.

### 2.3. Virus Production and Infection

To produce HIV_GR670_ virus, HEK293T cells at 7.0 × 10^6^ cells per 10 cm Cell Culture Dish (CELLTREAT, Ayer, MA, USA) coated with Poly L-lysine (Sigma, St. Louis, MO, USA) were co-transfected with pHIV_GR670_ at 13.00 µg and pSVIII-92US715.6 dual-tropic HIV-1 Env expression plasmid DNA (ATCC, Manassas, VA, USA) at 3.25 µg by PEI transfection method (Poly-sciences, Warrington, PA, USA). The culture medium was replaced with fresh D10 medium after 3 h for PEI transfections. Virus-containing supernatants were collected 48 h post-transfection. The supernatants were then filtered through a 0.45 µm sterile polyvinylidene difluoride (PVDF) membrane (CELLTREAT, Ayer, MA, USA) and concentrated up to 10-fold using Lenti-X Concentrator (TaKaRa, Shiga, Japan). The virus pellet was resuspended in 100 µL of complete medium containing 40 U/mL IL-2. HIV_GR670_ p24 at 50 µg, quantified using the HIV-1 Gag p24 DuoSet ELISA (R&D Systems, Minneapolis, MN, USA), was inoculated into 0.25 million human PBMCs that had been pre-stimulated with ImmunoCult Human CD3/CD28 T Cell Activator (STEMCELL Technologies, Vancouver, BC, Canada) for 3 days. Infection was carried out using spin infection in the presence of 10 µg/mL Polybrene (Sigma-Aldrich, Burlington, MA, USA) at 1000× *g* for 2 h at 30 °C. Following infection, cells were cultured in a 96-well U-bottom plate (CELLTREAT, Ayer, MA, USA).

### 2.4. Inhibitors

All inhibitors used in this study are listed in Table 1. The chemicals were initially resuspended in DMSO to prepare stock solutions at a concentration of 10.00 mM. These stock solutions were further diluted with R10 or complete medium to the appropriate optimized concentrations for cell treatment.

J-Lat cells were seeded in a 96-well U-bottom plate (CELLTREAT, Ayer, MA, USA) at a density of 0.25 million cells per well and treated with inhibitors at designated concentrations for 48 h in a CO_2_ incubator.

For human PBMCs infected with HIV_GR670_, the medium was replaced with complete medium containing inhibitors at designated concentrations 24 h post-infection, and the cells were incubated for an additional 48 h in a CO_2_ incubator.

### 2.5. Immunostaining

J-Lat Cells were stained with the Live/Dead Fixable Green Dead Cell Stain Kit, for 488 nm excitation (Invitrogen, Waltham, MA, USA). Human PBMCs isolated from bloods of healthy donors, purchased from BioIVT (Westbury, NY, USA), were prepared using the density gradient method with Lymphoprep (STEMCELL Technologies, Vancouver, BC, Canada). The cells were stained with Brilliant Violet 605-conjugated anti-human CD4 Mouse IgG2b (OKT4), Brilliant Violet 785-conjugated anti-human CD8 Mouse IgG1 (RPA-T8), and Alexa Fluor 700-conjugated anti-human CD3 Mouse IgG2a (OKT3) antibodies (BioLegend, San Diego, CA, USA). To assess cell viability, the Live/Dead Fixable Green Dead Cell Stain Kit, for 488 nm excitation (Invitrogen, Waltham, MA, USA), was included. Following staining, the cells were fixed using a final concentration of 2% methanol-free formaldehyde (Cell Signaling Technology, Danvers, MA, USA). All staining procedures were performed according to the corresponding manufacturer’s protocol.

### 2.6. NK-Cell Assays

K562 cells (0.5 million) were labeled with the CellTrace Far Red Cell Proliferation Kit for flow cytometry (CFSE) (Invitrogen, Waltham, MA, USA) for 10 min at 37 °C in a CO_2_ incubator, and then prepared at a concentration of 0.1 million cells per mL in complete medium with 100 U/mL human IL-2. A volume of 100 µL of human PBMCs at a concentration of 5.0 million cells per mL was co-cultured with an equal volume of CFSE-labeled K562 cells (50:1 effector to target ratio) for 4 h in the CO_2_ incubator. Following co-culture, the cells were stained with eBioscience Propidium Iodide Staining Solution (Invitrogen, Waltham, MA, USA) according to the manufacturer’s instructions.

### 2.7. Flow Cytometry

Immunostained cells were analyzed using the CytoFlex Flow Cytometer (Beckman Coulter, Brea, CA, USA). For J-Lat cell scanning, the ArC Amine Reactive Compensation Bead Kit for use with the Live/Dead Fixable Dead Cell Stain Kit (Invitrogen, Waltham, MA, USA) and GFP BrightComp eBeads Compensation Bead Kit (Invitrogen, Waltham, MA, USA) were utilized to calculate the compensation matrix. For scanning of human HIV_GR670_-infected PBMCs, in addition to the above compensation bead kits, the BD CompBeads Anti-Mouse Ig, k/Negative Control Compensation Particles Set (BD Biosciences, Franklin Lakes, NJ, USA) was used to optimize each signal and create the compensation matrix. Flow cytometry data analysis was performed using FlowJo v10.10.0 (FlowJo LLC, Ashland, OR, USA), and data plotting and statistical analysis were conducted using GraphPad Prism 10 (GraphPad Software, LLC, San Diego, CA, USA). All figures were designed using BioRender (https://www.biorender.com/, accessed on 14 February 2025, Toronto, ON, Canada).

## 3. Results

### 3.1. Latency Reversal Efficacy of β-Catenin Inhibitors in J-Lat Cell Line Models

β-catenin inhibition has been reported to release HIV-1 dormancy in J-Lat 8.4 cells [34]. Initially, we tested the latency-reversing efficacy of various β-catenin inhibitors, including some previously used, in J-Lat 8.4 cells to confirm their effects. We used seven different β-catenin inhibitors including BC21, PKF118-310 (PKF), PNU-74654 (PNU), ICRT-14, and ICRT-3, which target the inhibition of β-catenin and TCF/LEF transcription factor complex formation. ICG-001 (ICG) inhibits the interaction between β-catenin and CBP, which is a transcription coactivator for β-catenin and TCF/LEF transcription complexes. Adavivint (ADV) reduces total β-catenin protein levels [35]. PNU, ICG, and ADV have been used in previous studies [34].

The J-Lat 8.4 cells were treated with each inhibitor at two concentrations: BC21 (5.00 µM and 10.00 µM), PKF (0.50 µM and 1.00 µM), PNU (20.00 µM and 0.20 mM), ICRT-14 (25.00 µM and 50.00 µM), ICRT-3 (25.00 µM and 50.00 µM), ICG (10.00 µM and 20.00 µM), and ADV (20.00 nM and 0.20 µM). The HDAC inhibitor, vorinostat, at 10.00 nM was used as a positive control for latency reversion in J-Lat cells. Forty-eight hours after treatment, cells were stained with live/dead cell dye and reactivated cells were detected based on GFP signals using a flow cytometer. None of the β-catenin inhibitors demonstrated reactivation in J-Lat 8.4 cells (Figure 2A(I)). All tested concentrations of each drug did not impair cell viability (Figure 2A(II)).

We conducted additional experiments to explore the potential synergistic effect on latency reversal when combined with an HDAC inhibitor. This was prompted by reported synergistic interactions between the β-catenin inhibitor ADV and other LRAs [34]. PNU at a lower concentration (20.00 µM) slightly increased the reactivated cell population (median difference: 2.7) when combined with vorinostat, compared to treatment with vorinostat alone. However, this enhancement was not statistically significant (Figure 2B(I)). On the other hand, PKF, PNU, ICRT14, and ICG at higher concentrations (1.00 µM, 0.20 mM, 50.00 µM, and 20.00 µM, respectively) exhibited antagonistic effects with statistical significance, reducing the vorinostat-mediated latency reversal in J-Lat 8.4 cells (Figure 2B(I)). Consistent with panel A in Figure 2, none of the concentrations of each β-catenin inhibitor showed severe cytotoxic effects, even in combination with vorinostat (Figure 2B(II)).

The selected β-catenin inhibitors, BC21 and PKF, were further tested in other J-Lat cell lines, J-Lat 6.3 (Figure 2C,D), 9.2 (Figure 2E,F), and J-Lat 10.6 (Figure 2G,H), to determine the influence of proviral DNA integration sites. However, the β-catenin inhibitors tested in this study failed to reactivate any of the J-Lat cell lines on their own (Figure 2C(I),E(I),G(I)), and no synergistic effect was observed when combined with vorinostat (Figure 2D(I),F(I),H(I)). At any β-catenin inhibitor concentration, whether alone or in combination with vorinostat at 10.00 nM, no cytotoxic effect was observed in all tested J-Lat cell lines (Figure 2C(II)–H(II)). These data indicate that the β-catenin inhibitor, which prevents β-catenin and TCF/LEF transcription factor-mediated transcription, is not involved in HIV-1 latency.

### 3.2. Latency Reversal Efficacy of Tankyrase Inhibitors in J-Lat Cell Line Models

Tankyrase is an upstream enzyme that positively regulates β-catenin signaling by inhibiting the β-catenin destruction complex, thereby stabilizing β-catenin expression and promoting its translocation into the nucleus (Figure 1) [36]. To further investigate the involvement of β-catenin signaling in HIV-1 latency, we treated J-Lat 8.4 cells with two distinct tankyrase inhibitors, IWR-1-endo and XAV-939, which are known to inhibit downstream β-catenin signaling. The cells were treated with IWR-1-endo at concentrations of 0.10 µM and 1.00 µM, and XAV-939 at 0.01 µM and 0.10 µM, with or without 10.00 nM vorinostat, for 48 h. Reactivation was then detected using a flow cytometer. Similar to β-catenin inhibitors, tankyrase inhibitors themselves did not induce latency reversal in the J-Lat 8.4 cell line. (Figure 3A(I)). Interestingly, the latency reversal effect of vorinostat increased by an average of three-fold when combined with tankyrase inhibitors at both concentrations in J-Lat 8.4 cells (Figure 3B(I)). To confirm the effect of tankyrase inhibitors on vorinostat-mediated latency reversal in various J-Lat cell lines, we treated J-Lat 6.3 and 9.2 cells with IWR-1-endo and XAV-939 at the same concentrations used for J-Lat 8.4 cells. As J-Lat 10.6 cells are generally more reactive to LRAs than other J-Lat cell lines (Figure 2) [37], this subclone is unsuitable for assessing the synergistic latency reversal activity of the proposed combination treatment. Therefore, we excluded J-Lat 10.6 cells from evaluating the enhancement effect of tankyrase inhibitors on vorinostat-induced latency reversal. Consistent with the results in J-Lat 8.4 cells, neither tankyrase inhibitor alone reactivated HIV-1 latency in both J-Lat cell lines (Figure 3C(I),E(I)). However, both inhibitors significantly enhanced vorinostat-mediated latency reversal in all tested cell lines (Figure 3D(I),F(I)). These findings strongly suggest that tankyrase enzymatic activity reduces the efficacy of vorinostat-mediated latency reversal in HIV-1 latently infected cells. Moreover, this enhancement effect was not affected by the proviral DNA integration sites. Therefore, we selected J-Lat 8.4 cells as the representative cell line for further experiments. No cytotoxicity was observed at any concentration of tankyrase inhibitors, either alone or in combination with vorinostat at 10.00 nM (Figure 3(II)).

### 3.3. Latency Reversal Efficacy of Hippo Inhibitors in J-Lat Cell Line Models

The downstream signaling of tankyrase involves not only β-catenin signaling but also Hippo signaling [23]. To further characterize the specific downstream signaling pathways regulated by tankyrase that influence vorinostat-mediated latency reversal efficacy, we assessed the effects of combining Hippo pathway inhibitors with vorinostat in J-Lat 8.4 cells (Figure 4).

We used four different Hippo pathway inhibitors: Peptide-17, TED-347, and verteporfin (VET), which inhibit the YAP-TEAD protein interaction, and CA3 (CIL56), which inhibits YAP-TEAD transcriptional activity. The cells were treated for 48 h with Peptide-17 at 0.25 nM and 2.50 nM, TED-347 at 5.00 nM and 50.00 nM, CA3 (CIL56) at 1.00 nM and 10.00 nM, and VET at 5.00 nM and 50.00 nM before flow cytometry analysis. While none of the Hippo inhibitors exhibit latency reversal activity on their own in J-Lat 8.4 cells (Figure 4A(I)), consistent with the effect observed for tankyrase inhibitors in Figure 3, all four inhibitors enhance the vorinostat-mediated latency reversal at both concentrations to a similar extent as the tankyrase inhibitors (Figure 4B(I)). These data reveal that the Hippo pathway suppresses the potential efficacy of vorinostat-induced latency reversal, and inhibition of the Hippo pathway by tankyrase inhibitors is crucial for enhancing vorinostat-mediated latency reversal. No cytotoxic effects were observed at any concentration (Figure 4(II)).

### 3.4. Latency Reversal Efficacy of PARP Inhibitors in J-Lat Cell Line Models

Tankyrase is classified as a member of the PARP superfamily, specifically PARP5. While PARP inhibitors were originally developed to target PARP1, PARP2, and PARP3, all four FDA-approved PARP inhibitors have been shown to inhibit tankyrase activity [38]. To investigate their potential role in enhancing vorinostat-mediated latency reversal, we tested four FDA-approved PARP inhibitors: olaparib, rucaparib, niraparib, and talazoparib. J-Lat 8.4 cells were treated with three different concentrations of each PARP inhibitor, olaparib at 2.00 nM, 20.00 nM, and 0.20 µM; rucaparib at 1.00 nM, 10.00 nM, and 0.10 µM; niraparib at 5.00 nM, 50.00 nM, and 0.50 µM; and talazoparib at 0.50 nM, 5.00 nM, and 50.00 nM, for 48 h. The reactivated cell population was detected using GFP signals via flow cytometry. Similar to the effects observed with tankyrase inhibitors (Figure 3) and Hippo inhibitors (Figure 4), none of the PARP inhibitors alone exhibited latency reversal activity in J-Lat 8.4 cells (Figure 5A(I)). However, when combined with vorinostat, all four PARP inhibitors increased reactivated cell populations as effectively as, or in some cases, better than, tankyrase and Hippo inhibitors (Figure 5B(I)). None of the PARP inhibitors, at any tested concentration in combination with 10.00 nM vorinostat, exhibited severe cytotoxic effects in J-Lat 8.4 cells (Figure 5(II)). These data indicate that PARP inhibitors enhance the potential of vorinostat-mediated latency reversal by inhibiting Hippo pathway signaling through the suppression of tankyrase activity.

### 3.5. Latency Reversal Efficacy of PARP Inhibitor in Human Primary Cell Model and Immune Activation

To further confirm the enhancement effect of PARP inhibitors on vorinostat-mediated latency reversal in more physiologically and clinically relevant models using human primary cells, we developed a dual-fluorescent HIV-1 pseudotyped construct, HIV_GR670_, which harbors GFP and miRFP670nano3 fluorescent genes in place of the nef gene. This construct enables the distinction between actively and latently infected cell populations via flow cytometry. The GFP gene is transcribed by the HIV-1 LTR promoter, marking actively infected cells (Figure 6A), while the miRFP670nano3 gene is transcribed by an independent constitutively active promoter, EF-1α. As a result, cells positive for miRFP670nano3 alone without GFP signal are identified as latently infected cell populations (Figure 6A). We isolated human PBMCs from healthy donors and stimulated them with CD3/CD28 T-cell activation cocktails for three days, followed by infection using the spin-inoculation method with the pseudotyped HIV_GR670_ virus harboring an HIV-1 dual-tropic Env. Twenty-four hours post-infection, infected PBMCs were treated with vorinostat at 0.50 nM and talazoparib, one of the four PARP inhibitors used in this study, at 5.00 nM and cultured for an additional 48 h. Following 48 h of post-treatment, the cells were immunostained with anti-CD3, CD4, and CD8 antibodies for flow cytometry analysis (Figure 6A). The actively infected CD4 T-cell population was statistically increased when treated with vorinostat (Figure 6B), while the latently infected cell population was correspondingly reduced (Figure 6C). Latently infected CD4 T-cell populations were further reduced with the combination of vorinostat and talazoparib treatment, despite talazoparib alone having no effect on latently infected cell populations (Figure 6C), which was consistent with the data observed in J-Lat 8.4 cell models (Figure 5). Interestingly, the increased number of actively infected CD4 T-cell populations by vorinostat treatment was diminished in combination with talazoparib treatment (Figure 6B), even though the latently infected cell populations were further reduced by the combination treatment (Figure 6C). In addition, actively infected cell populations were drastically reduced almost to a half of the original number by the talazoparib-only treatment (Figure 6B). This suggests that talazoparib may activate immune cells, particularly NK cells in this case, due to the use of human PBMCs isolated from HIV-negative donors, in which anti-HIV CD8 T cells are not present.

### 3.6. Enhancement of NK-Cell Cytotoxic Activity by PARP Inhibitor

HIV-1 active infection has been reported to cause significant downregulation of surface HLA-E levels [39], which serves as a marker for NK-cell targeting. These activated NK cells then kill the actively infected cell populations. To test this hypothesis, we performed an in vitro NK-cell assay. The viability of K562 cells, lymphoblasts used as target cells in NK-cell-mediated cytotoxicity, was measured using propidium iodide for live/dead cell staining in flow cytometry, following the published protocol [40]. Human PBMCs were stimulated with CD3/CD28 T-cell stimulation cocktails for three days and then treated with talazoparib at three different concentrations, 0.50 nM, 5.00 nM, and 50.00 nM, for 48 h before coculturing with K562 cells labeled with CFSE (Figure 6D). We observed that the viability of K562 cells cocultured with talazoparib-treated PBMCs was dose-dependently reduced at maximum 10% compared to those cocultured with untreated PBMCs (Figure 6E). While this does not provide direct evidence that talazoparib-treated NK cells kill HIV actively infected CD4 T cells, this data suggests that talazoparib enhances the latency reversal effect of vorinostat in HIV-1 latently infected human CD4 T cells, subsequently facilitating the elimination of reactivated cell populations through NK-cell activation.

## 4. Discussion

While the “Kick and Kill” strategy remains a cornerstone in the pursuit of an HIV cure, the research presented underscores the need for innovative approaches to overcome the limitations of current LRAs. The synergistic effects observed between vorinostat, the first FDA-approved HDAC inhibitor, and PARP inhibitors represent a significant advancement in addressing the challenges of reactivating and subsequently eliminating latently infected cells. Results from J-Lat cell line models and primary cell experiments demonstrate that while PARP inhibitors alone lack latency reversal efficacy, their combination with vorinostat substantially enhances latency reversal activity (Figure 5 and Figure 6). This synergy was consistent across cell line models and primary cells, with the combination therapy achieving a three-fold increase in latency reversal in J-Lat cells and a further reduction in latently infected primary CD4 T cells compared to vorinostat alone. Furthermore, the ability of talazoparib to reduce actively infected cell populations suggests a novel role in enhancing immune-mediated cytotoxicity, particularly through the activation of NK cells (Figure 6). These findings demonstrate the dual advantages of this combination therapy as it not only targets the latent reservoir but also enhances the capacity of immune systems to eliminate reactivated HIV-infected cells.

PARP inhibitors, originally developed for targeting DNA repair mechanisms in cancer therapy, have shown remarkable efficacy in the treatment of ovarian and breast cancers [41,42]. Specifically, these inhibitors exploit the concept of synthetic lethality in tumors with defective homologous recombination repair pathways, such as those harboring BRCA1 or BRCA2 mutations [43,44]. By inhibiting PARP enzymes, which play a critical role in single-strand DNA break repair, these drugs exacerbate genomic instability in cancer cells, ultimately leading to cell death. This mechanism has been validated in clinical settings, with FDA-approved PARP inhibitors including olaparib, rucaparib, niraparib, and talazoparib demonstrating significant survival benefits for patients with advanced ovarian and breast cancers [41]. Their application in oncology has also revealed their capacity to modulate immune responses. PARP inhibitors have been shown to enhance NK-cell-mediated antitumor immunity by inducing the expression of NKG2D ligands on leukemic stem cells, thereby sensitizing them to NK-cell-mediated elimination [45,46,47,48,49]. Moreover, PARP inhibitors influence the production of cytokines, including IFN-γ and TNF-α, which are crucial for recruiting NK cells and promoting their cytopathic activity [50]. Therefore, these dual functionalities, enhancing HDAC inhibitor-mediated latency reversal effects and providing immune stimulation, highlight the versatility of PARP inhibitors as therapeutic agents and their potential to be repurposed for HIV cure strategies.

Although PARP inhibitors primarily target PARP1, PARP2, and PARP3, FDA-approved PARP inhibitors also inhibit PARP5 (tankyrase) enzymatic activity [38]. Notably, IWR-1-endo, a specific selective inhibitor of tankyrase among the 17 PARP subfamily proteins [38], also enhanced the latency reversal efficacy of vorinostat to a similar extent as PARP inhibitors (Figure 3 and Figure 5). This suggests that the enhancement effect of PARP inhibitors is primarily mediated through tankyrase inhibition. The mechanistic insights provided by this study further elucidate the pathways involved in latency reversal. The inhibition of tankyrase and its downstream signaling pathway, Hippo signaling, emerged as critical mediators of HDAC inhibitor-mediated latency reversal efficacy. By targeting this pathway, PARP inhibitors effectively suppress tankyrase activity, thereby enhancing the performance of HDAC inhibitors in latency reversion. Tankyrase stabilizes and relocates yes-associated protein (YAP) into the nucleus to induce Hippo signaling via promoting polyADP-ribosylation of angiomotin (AMOT), and promoting its degradation through the ubiquitin–proteasomal pathway by E3 ligase RNF146 [23] (Figure 1). Hippo signaling is primarily mediated by a potent transcription coactivator, YAP, which has been defined to have essential roles in organ size control, tissue regeneration, and self-renewal. YAP binds to the TEAD transcription factor and modulates transcription at several levels, including transcriptional regulation, association of enhancers, and interaction with chromatin-regulating proteins to control gene expression and chromatin accessibility by other transcription factors [25,26,27]. AMOT proteins have been shown to inhibit YAP nuclear translocation through its phosphorylation and degradation (Figure 1) [51].

Although Hippo signaling has recently been reported to enhance innate immunity by upregulating type-I interferon and cytokines in response to inhibiting HIV infection [52], there are no reports examining viral gene silencing. This research is the first to indicate the involvement of Hippo signaling in HIV-1 silencing. Multiple studies have suggested that bromodomain-containing protein 4 (BRD4), a member of the bromodomain and extraterminal (BET) protein family, plays a role in the regulation of HIV transcription and latency [53,54,55]. In support of this, inhibition of BRD4/BET by the pan-BET inhibitor JQ1 has been shown to activate HIV transcription [56]. BRD4 can be recruited to the HIV promoter through binding to different acetyl-histones, including acetyl-histone H3 and acetyl-histone H4 [56]. BRD4 is reported as an inhibitory factor that strongly suppresses Tat-transactivation in latently infected T-cell lines, and that antagonizing this restriction by JQ1 could reactivate HIV-1 latency [54]. YAP interacts with BRD4 and contributes to transcriptional dependency in cancer cells by recruiting BRD4 to specific genomic regions that mediate BRD4-regulated gene expression [27]. Therefore, it is hypothesized that YAP may also be involved in HIV-1 LTR silencing by recruiting BRD4 to acetylated histones induced by HDAC inhibitors. PARP inhibitors reduce tankyrase activity, leading to YAP degradation and consequently impairing BRD4 recruitment to the HIV-1 LTR. This represents one of the hypothesized molecular mechanisms underlying the PARP inhibitor-mediated enhancement of HIV-1 latency reversion induced by HDAC inhibitors. Further research is necessary to elucidate the detailed molecular mechanisms.

Tankyrase also induces polyADP-ribosylation of AXIN1/2, an essential central scaffolding protein in the β-catenin destruction complex, and targets it for degradation, which adjusts the responsiveness of cells to Wnt signals (Figure 1) [57,58]. The Wnt/β-catenin signaling is a major transduction system in eukaryotic cells [59]. β-catenin is hypophosphorylated in the cytoplasm by the β-catenin destruction complex, following degradation via the ubiquitin–proteasomal pathway (Figure 1). When the Wnt signal is induced, β-catenin is translocated into the nucleus, where it associates with transcription factors TCF/LEF family proteins that are the endpoint effector of the Wnt signaling and drive the transcription of target genes. TCF/LEF interacts with the DNA and recruits a β-catenin cofactor to regulate gene expression. It regulates hundreds of genes involved in cell communication, survival, differentiation, and physiologic functions [28,29]. Wnt/β-catenin signaling was reported to suppress the immune response to cancers and pathogens [60,61,62,63], indicating that Wnt/β-catenin signaling would negatively affect the immune system. Several effects of Wnt/β-catenin signaling on T-cell differentiation have previously been reported. Activation of Wnt/β-catenin signaling during CD4 and CD8 T-cell priming inhibits effector differentiation due to promoting the generation of self-renewing central memory T cells (T_CM_) [64,65,66]. It has also been demonstrated that Wnt/β-catenin signaling promotes T helper (Th) 2 over Th1 polarization and enhances the survival of naturally occurring regulatory T cells [67]. When β-catenin is impaired to interact with coactivator proteins to form a transcriptional complex with TCF/LEF proteins in the nucleus, T_CM_ is promoted to be differentiated to effector memory subset (T_EM_), which is a terminally differentiated phenotype and plays a significant role in immunity against pathogenic agents. Furthermore, active Wnt/β-catenin signaling is known to encourage immune tolerance, especially in NK-cell-mediated responses [60,61,62,63,66,68], suggesting that inhibiting this signaling pathway could also enhance immune cell function.

Recent research indicated that inhibiting Wnt/β-catenin signaling could lead to the reactivation of latently infected cells [34]. Consequently, Wnt/β-catenin inhibition was predicted to be a promising candidate for the development of next-generation LRA therapeutic strategy. These therapies possess the potential to reactivate HIV-infected dormant cells and subsequently stimulate immune functions, particularly those of NK cells, to eradicate HIV reservoirs. Although β-catenin signaling inhibition has not been directly demonstrated to influence HIV-1 latency reversion in our experiments (Figure 2), the enzymatic suppression of tankyrase by PARP inhibitors may potentially enhance immune function by modulating β-catenin signaling. Improvements in latency reversal efficacy and the observed reductions in reservoir size underscore the therapeutic potential of PARP inhibitors as part of a combination strategy.

Although these findings represent significant progress, it is essential to address the limitations and challenges that remain. The synergistic effects of PARP inhibitors with other HDAC inhibitors, such as Romidepsin or Panobinostat, require comprehensive confirmation. Furthermore, in addition to HDAC inhibitors as combination LRA partners, additional LRAs, such as JQ1, the PKC agonist, IAPi/SMACm or TLR9 antagonist, should be evaluated in combination with PARP inhibitors. The immunomodulatory effects of PARP inhibitors on broader immune cell populations require further investigation, particularly to assess their impact on T-cell responses and overall immune homeostasis. The in vitro findings must be validated in in vivo models to confirm the safety and efficacy of this combination therapy. As the FDA has approved both HDAC and PARP inhibitors for cancer treatment, this pre-existing approval streamlines the pathway for clinical trials in the context of HIV cure strategies. By leveraging these established therapeutics, the timeline for translating laboratory findings into patient care could be significantly accelerated. The broader implications of enhancing immune-mediated clearance mechanisms, as observed with NK-cell activation, suggest potential synergies with other immunotherapeutic approaches. For instance, the combination of latency reversal with immune checkpoint inhibitors or therapeutic vaccines could amplify the therapeutic outcomes, offering a multi-pronged approach to reservoir clearance. Such strategies could address the heterogeneity of HIV reservoirs and the diverse pathways maintaining latency.

In conclusion, the combination of HDAC inhibitors and PARP inhibitors represents a compelling advancement in HIV cure research. By enhancing latency reversal and immune-mediated clearance, this approach addresses key barriers to achieving a functional cure. The integration of these findings into clinical studies may pave the way for transformative advancements in the fight against HIV. Continued research and development in this area is essential to fully realize the potential of combination therapies, ultimately bringing us closer to an effective and accessible cure for HIV.

## Figures and Tables

**Figure 1 viruses-17-00400-f001:**
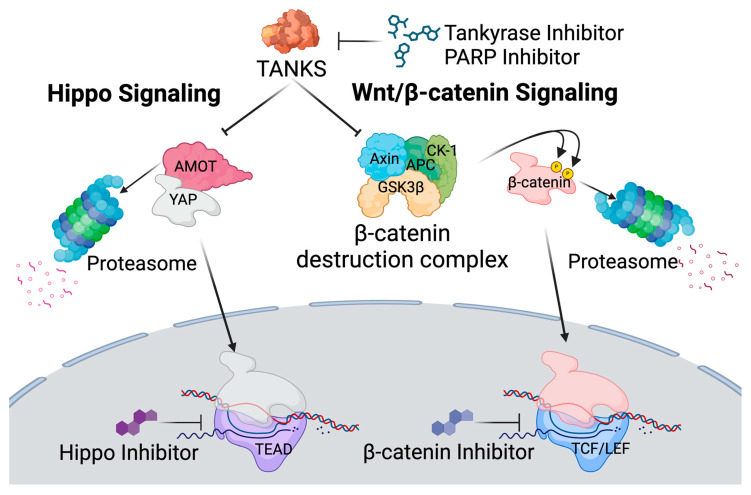
Schematic representation of tankyrase downstream pathways.

**Figure 2 viruses-17-00400-f002:**
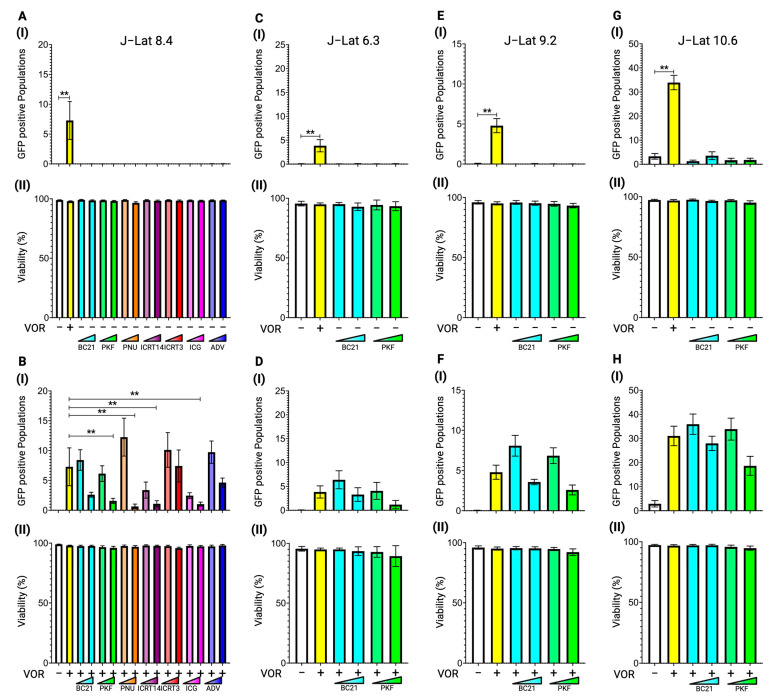
Latency reversal effect of β-catenin inhibitors in J-Lat cell lines. (**A**) (**I**) The latency reversal effect of β-catenin inhibitors in J-Lat 8.4 cells and (**II**) cell viability, determined by staining with live/dead staining dye, were assessed by flow cytometry. An HDAC inhibitor, vorinostat, at 10.00 nM was used as the positive control. The J-Lat 8.4 cells were treated with seven different β-catenin inhibitors at two different concentrations: BC21 at 5.00 µM or 10.00 µM, PKF118-310 (PKF) at 0.50 µM or 1.00 µM, ICG-001 (ICG) at 10.00 µM or 20.00 µM, PNU-74654 (PNU) at 20.00 µM or 0.20 mM, ICRT-14 at 25.00 µM or 50.00 µM, ICRT-3 at 25.00 µM or 50.00 µM, or Advivint (ADV) at 20.00 nM or 0.20 µM. (**B**) (**I**) The synergistic latency reversal effect of β-catenin inhibitors in combination with vorinostat and (**II**) cell viability in J-Lat 8.4 cells were assessed by flow cytometry. The concentrations of each β-catenin inhibitor and vorinostat were the same as in (**A**). The same experiments were conducted in (**C**,**D**) J-Lat 6.3, (**E**,**F**) J-Lat 9.2, and (**G**,**H**) J-Lat 10.6 cells. Statistical significance was determined using (**I**) the Mann–Whitney U test with the relative differences in GFP-positive cell populations and (**II**) the Wilcoxon matched-pairs signed rank test compared with (**A**,**C**,**E**,**G**) the DMSO treated negative control sample or (**B**,**D**,**F**,**H**) the vorinostat-only treated sample. Error bars represent the standard error from five independent experiments, and ** *p*-value < 0.01 with more than two-fold median differences observed.

**Figure 3 viruses-17-00400-f003:**
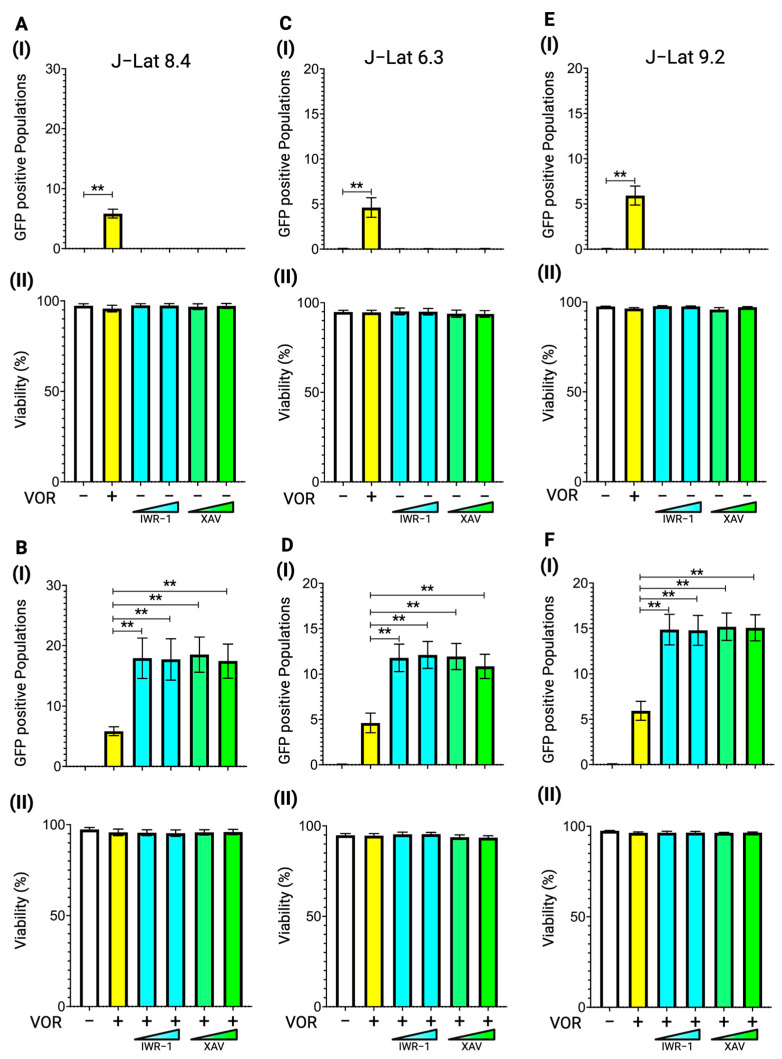
Latency reversal effect of tankyrase inhibitors in J-Lat cell lines. (**A**) (**I**) The latency reversal effect of tankyrase inhibitors in J-Lat 8.4 cells and (**II**) cell viability, determined by staining with live/dead staining dye, were assessed by flow cytometry. An HDAC inhibitor, vorinostat, at 10.00 nM was used as the positive control. The J-Lat 8.4 cells were treated with two different tankyrase inhibitors at two different concentrations: IWR-1-endo at 0.10 µM or 1.00 µM or XAV-939 at 10.00 nM or 0.10 µM (**B**) (**I**) The synergistic latency reversal effect of tankyrase inhibitors in combination with vorinostat and (**II**) cell viability in J-Lat 8.4 cells were assessed by flow cytometry. Concentrations of each tankyrase inhibitor and vorinostat remain consistent with prior experiments (**A**). The same experiments were conducted in (**C**,**D**) J-Lat 6.3 and (**E**,**F**) J-Lat 9.2 cells. Statistical significance was determined using (**I**) the Mann–Whitney U test with the relative differences in GFP-positive cell populations and (**II**) the Wilcoxon matched-pairs signed rank test compared with (**A**,**C**,**E**) the DMSO treated negative control sample or (**B**,**D**,**F**) the vorinostat-only treated sample. Error bars represent the standard error from five independent experiments, and ** *p*-value < 0.01 with more than two-fold median differences observed.

**Figure 4 viruses-17-00400-f004:**
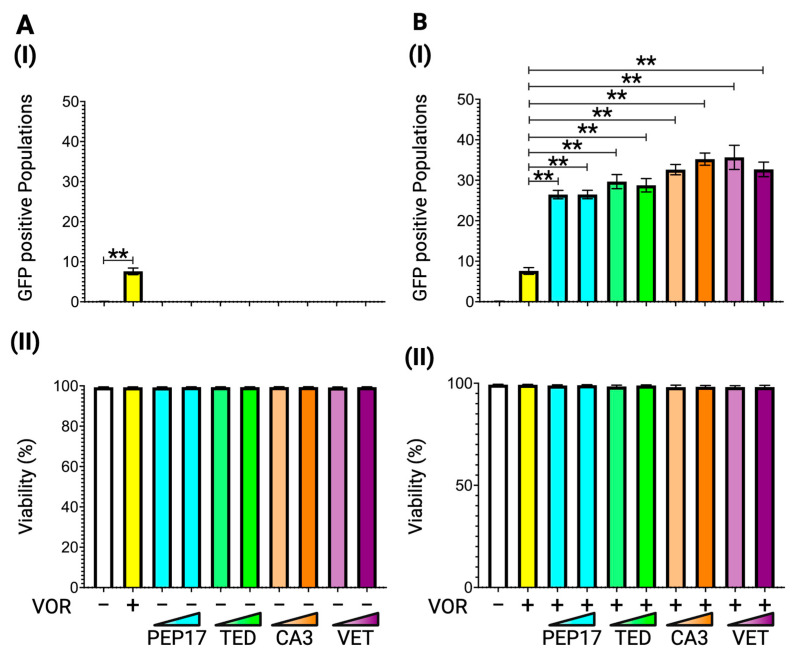
Latency reversal effect of Hippo inhibitors in J-Lat cell lines. (**A**) (**I**) The latency reversal effect of Hippo inhibitors in J-Lat 8.4 cells and (**II**) cell viability, determined by staining with live/dead staining dye, were assessed by flow cytometry. An HDAC inhibitor, vorinostat, at 10.00 nM, was used as the positive control. The J-Lat 8.4 cells were treated with four Hippo inhibitors at two different concentrations: peptide-17 at 0.25 nM or 2.50 nM, TED-347 at 5.00 nM or 50.00 nM, CA3 (CIL56) at 1.00 nM or 5.00 nM, and verteporifin (VET) at 0.50 nM or 5.00 nM. (**B**) (**I**) The synergistic latency reversal effect of Hippo inhibitors in combination with vorinostat and (**II**) cell viability in J-Lat 8.4 were assessed by flow cytometry. Concentrations of each Hippo inhibitor and vorinostat remain consistent with the prior experiments (**A**). Statistical significance was determined using (**I**) the Mann–Whitney U test with the relative differences in GFP-positive cell populations and (**II**) the Wilcoxon matched-pairs signed rank test compared with (**A**) the DMSO treated negative control sample or (**B**) the vorinoatat-only treated sample. Error bars represent the standard error from five independent experiments, and ** *p*-value < 0.01 with more than two-fold median differences observed.

**Figure 5 viruses-17-00400-f005:**
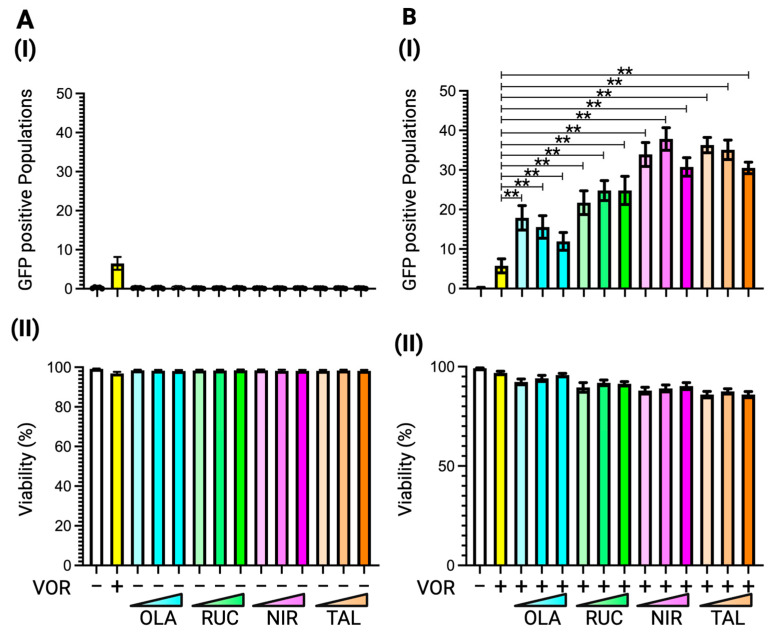
Latency reversal effect of PARP inhibitors in J-Lat 8.4 cells. (**A**) (**I**) The latency reversal effect of PARP inhibitors in J-Lat 8.4 cells and (**II**) cell viability, determined by staining with live/dead staining dye, were assessed by flow cytometry. An HDAC inhibitor, vorinostat, at 10.00 nM, was used as the positive control. The J-Lat 8.4 cells were treated with four FDA-approved PARP inhibitors at three different concentrations: olaparib at 2.00 nM, 0.02 µM, or 0.20 µM; rucaparib at 1.00 nM, 10.00 nM, or 0.10 µM; niraparib at 5.00 nM, 50.00 nM, or 0.50 µM; and talazoparib at 0.50 nM, 5.00 nM, or 50.00 nM. (**B**) (**I**) The synergistic latency reversal effect of PARP inhibitors in combination with vorinostat and (**II**) cell viability in J-Lat 8.4 cells were assessed by flow cytometry. Concentrations of each PARP inhibitor and vorinostat remain consistent with the prior experiments (**A**). Statistical significance was determined using (**I**) the Mann–Whitney U test with the relative differences in GFP-positive cell populations and (**II**) the Wilcoxon matched-pairs signed rank test compared with (**A**) the DMSO treated negative control sample or (**B**) the vorinoatat-only treated sample. Error bars represent the standard error from five independent experiments, and ** *p*-value < 0.01 with more than two-fold median differences observed.

**Figure 6 viruses-17-00400-f006:**
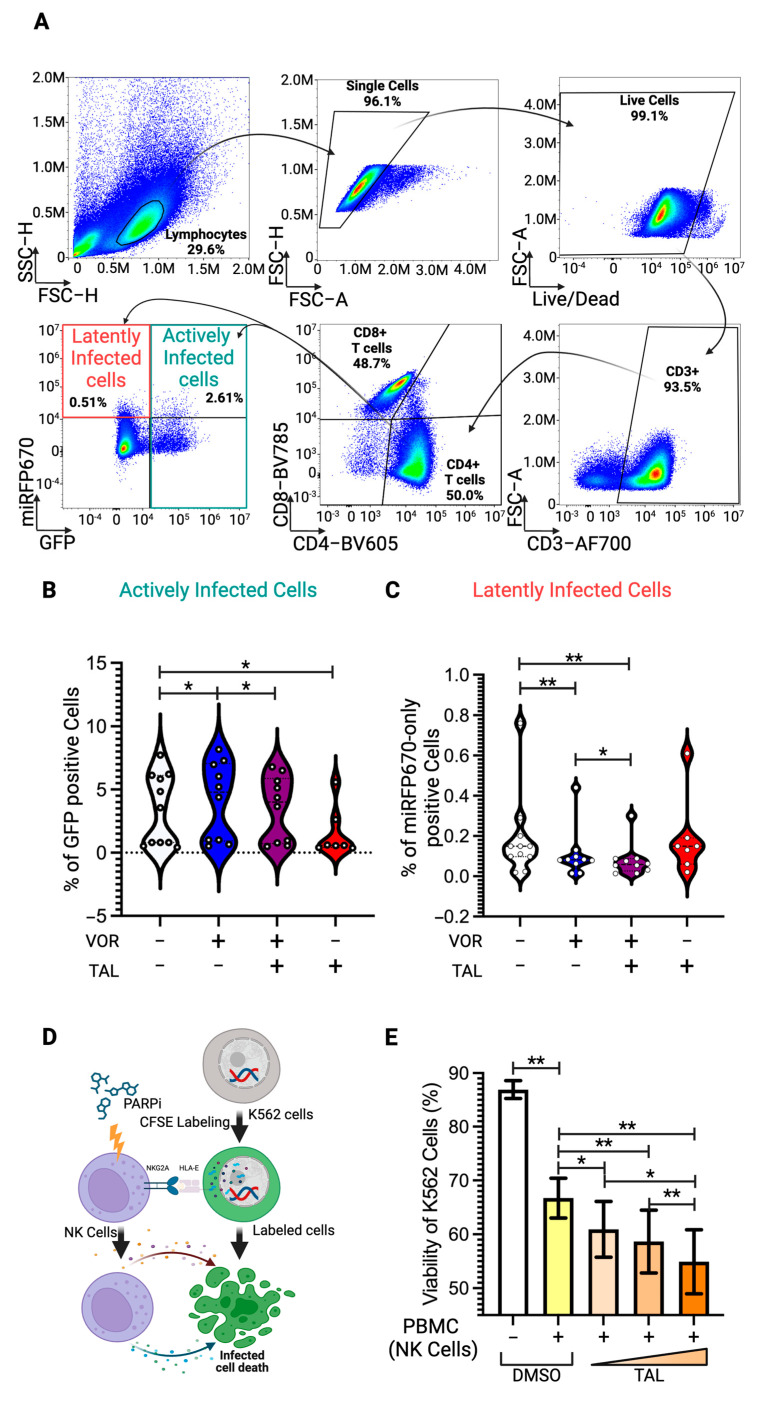
Latent reservoir reduction and NK-cell activation by PARP inhibitor. (**A**) Gating strategy of HIV_GR670_-infected human PBMCs treated with vorinostat at 0.50 nM and/or talazoparib at 0.05 µM that were immunostained with anti-CD3-AF700, CD4-BV605, and CD8-BV785 antibodies for flow cytometry analysis. Lymphocyte populations were first identified using FSC-H and SSC-H parameters, followed by the selection of single cells using FSC-A and FSC-H parameters. Early-stage dead cells were excluded by staining with a live/dead dye, and only live cells were further analyzed. Infected CD4 T cells were characterized by the CD3+, CD8−, and CD4+ cell subset. Actively infected cells were identified as the GFP-positive populations and latently infected cells were determined as the miRFP670-only positive cells without a GFP signal. Violin plot shows (**B**) the percentage of GFP-positive cells (actively infected cell population) and (**C**) the percentage of miRFP670-only positive cells (latently infected cell population) with or without vorinostat (VOR) and/or talazoparib (TAL) treatment from ten independent experiments of vorinostat-only and vorinostat/talazoparib-treated samples, and seven independent experiments of talazoparib-only treated samples. Statistical significance was determined using the Wilcoxon matched-pairs signed rank test, comparing untreated controls with other samples, and vorinostat-only treated samples with vorinostat and talazoparib double-treated samples. (**D**) Schematic representation of the in vitro NK-cell assay. K562 cells were pre-stained with CFSE before coculturing with PARP inhibitor-stimulated PBMCs containing NK-cell populations. K562 cells were identified as CFSE-positive cell populations by flow cytometry. (**E**) K562 cell viability, detected by propidium iodide staining, was plotted. Statistical significance was determined using the Wilcoxon matched-pairs signed rank test, comparing K562 cell viability with and without DMSO-treated PBMCs and DMSO-treated PBMCs versus PBMCs treated with three different concentrations of talazoparib (0.50 nM, 0.05 µM, and 0.50 µM). Error bars represent the standard error from eight independent experiments and * *p*-value < 0.05 and ** *p*-value < 0.01.

**Table 1 viruses-17-00400-t001:** Chemical information.

Name	Catalog Number	Vendor
**HDAC Inhibitor**		
Vorinostat	SML0061	Sigma-Aldrich
**ß-catenin Inhibitor**		
BC21	219334	Sigma-Aldrich
PKF118-310	219331	Sigma-Aldrich
PNU-74654	S8429	Selleckchem
ICRT14	SML0203	Sigma-Aldrich
iCRT3	219332	Sigma-Aldrich
ICG-001	S2662	Selleckchem
Adavivint	SM04690	Sigma-Aldrich
**Tankyrase Inhibitor**		
IWR-1-endo	681669	Sigma-Aldrich
XAV-939	S1180	Sellecchem
**Hippo Inhibitor**		
Peptide-17	S8164	Selleckchem
TED-347	S8951	Selleckchem
Verteporfin	S1786	Selleckchem
CA3 (CIL56)	S8661	Selleckchem
**PARP Inhibitor**		
Olaparib	S1060	Selleckchem
Rucaparib	S4948	Selleckchem
Niraparib	S2741	Selleckchem
Talazoparib	S7048	Selleckchem

## Data Availability

Data is contained within the article.

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
