# Peer review of "Targeting Latent HIV Reservoirs: Effectiveness of Combination Therapy with HDAC and PARP Inhibitors"

_viruses, 2025, doi:10.3390/v17030400_

Round 1
Reviewer 1 Report
Comments and Suggestions for Authors
In the work “Targeting Latent HIV Reservoirs: Effectiveness of Combination 2 Therapy with HDAC and PARP Inhibitors” by Hasset Tibebe and colleagues, the combination of an HDACi like vorinostat and PARP inhibitors is proposed as a novel treatment option to reverse HIV-1 latency and to eliminate latently infected cells reactivating, by NK cell activity.
It is an interesting work but several issues need to be fixed before publication.
1) In Table 1, vorinostat is included among bcatenin inhibitors, is this a mistake or authors have indications that vorinostat possesses also this alternative function?
2) At page 6 line 178 authors stated: “…which target the inhibition of β-catenin and TCF/LEF transcription factor complex formation”. The TCF/LEF transcription factor complex formation is not introduced previously in the introduction section in relation to β-catenin, and it is not described in the result section paragraph where it appears for the first time (it is only depicted in Figure1 but not mentioned in the figure description. Therefore, authors should clarify TCF/LEF role in relation to β-catenin and the different inhibitors used. Moreover, it seems that authors are describing a bcatenin activation function of bcatenin inhibitors: “…which target the inhibition of β-catenin…”
3) the % of GFP positive cells after Vorinostat treatment in JLat 10.6 cells seems to be higher (above 30%) than expected at such a low concentration (10 nM), even despite the higher capability of these cell clone to reactivate from HIV-1 transcription from latency following LRA treatments. The authors should provide a detailed procedure of the growth of these cells before the treatment and how many cells were stimulated and at what cell density (these information is not present in the materials and methods section, nor it is present in the description of the figure).
4) Authors should show data from all JLat cell lines for each set of experiments, and not reducing the number of JLat clones analyzed going from bcatenin to PARP inhibitors. As an alternative, authors should justify their choice.
5) Figure 6 is not well described: it is not clear what is represented in panel A; panels B and C graphical view is difficult to interpret, I suggest a different type of graphic.
6) The conclusion of the result section 3.6 “These data indicate that talazoparib increases the latency reversal effect of vorinostat in HIV-1 latently infected human CD4 T-cells and subsequently eliminates reactivated cell populations through the activation of NK cell activity.” is only partially sustained by data. Authors have shown an increased NK activity upon treatment of PBMCs with talazoparib and vorinostat, against K562 cells, not against reactivated cell population.
Author Response
We sincerely appreciate your reviewing our manuscript and providing valuable comments. Below is our response to each comment.
Reviewer Comment 1: In Table 1, vorinostat is included among bcatenin inhibitors, is this a mistake or authors have indications that vorinostat possesses also this alternative function?
Authors Response: Thank you for pointing out the mistake in Table 1. We have corrected it accordingly.
Reviewer Comment 2.1: At page 6 line 178 authors stated: “…which target the inhibition of β-catenin and TCF/LEF transcription factor complex formation”. The TCF/LEF transcription factor complex formation is not introduced previously in the introduction section in relation to β-catenin, and it is not described in the result section paragraph where it appears for the first time (it is only depicted in Figure1 but not mentioned in the figure description. Therefore, authors should clarify TCF/LEF role in relation to β-catenin and the different inhibitors used.
Authors Response: In the original manuscript, we described the β-catenin signaling pathway in the Discussion section in lines 513–518 on page 18. We have highlighted this sentence in red in the revised manuscript. Additionally, in response to the reviewer’s suggestion, we have also briefly introduced the Hippo and β-catenin signaling pathways in the Introduction section in lines 73–79 on page 2.
Reviewer Comment 2.2: Moreover, it seems that authors are describing a bcatenin activation function of bcatenin inhibitors: “…which target the inhibition of β-catenin…”
Authors Response 2.2: In this manuscript, we do not discuss β-catenin activation function in the context of β-catenin inhibitor treatment.
Reviewer Comment 3: the % of GFP positive cells after Vorinostat treatment in JLat 10.6 cells seems to be higher (above 30%) than expected at such a low concentration (10 nM), even despite the higher capability of these cell clone to reactivate from HIV-1 transcription from latency following LRA treatments. The authors should provide a detailed procedure of the growth of these cells before the treatment and how many cells were stimulated and at what cell density (these information is not present in the materials and methods section, nor it is present in the description of the figure).
Authors Response 3: As the reviewer mentioned, J-Lat 10.6 cells are generally considered more reactive than other J-Lat cell lines when exposed to LRAs (doi: 10.3389/fimmu.2021.682182). As suggested by the reviewer, we have added more details to the 2. Materials and Methods section to provide a comprehensive guide to our cell culture procedures. Those additional details have been highlighted in red in the revised manuscript.
Reviewer Comment 4: Authors should show data from all JLat cell lines for each set of experiments, and not reducing the number of JLat clones analyzed going from bcatenin to PARP inhibitors. As an alternative, authors should justify their choice.
Authors Response 4: As the reviewer mentioned and in response to above comment, J-Lat 10.6 cells are generally more reactive to LRAs compared to other J-Lat cell lines, making them unsuitable for evaluating the synergistic latency reversal activity of combination therapy. Consequently, we excluded J-Lat 10.6 cells from Figure 3. This has been further explained on page 8, between lines 263 and 267.
Figure 3 shows that Tankyrase inhibitors universally enhance vorinostat-mediated latency reversal in all three J-Lat cell lines, J-Lat 8.4, 6.3, and 9.2. Based on this, we selected J-Lat 8.4 cells as a representative cell line for further characterization of Hippo and PARP inhibitors' functions. This was also explained on pages 8 and 9, lines 273 and 274.
Reviewer Comment 5: Figure 6 is not well described: it is not clear what is represented in panel A; panels B and C graphical view is difficult to interpret, I suggest a different type of graphic.
Authors Response 5: The units of the x-axis and y-axis in each flow panel of Figure 6A have been revised to a larger font size and also include the conjugated fluorophores for each antibody. In addition, we have further included the lymphocytes, single-cell, and live-cell gating strategies in Figure 6A. The gating strategy for latently infected and actively infected cells in Figure 6A has been aligned with Figures 6B and 6C using the corresponding colors (green for actively infected cells and red for latently infected cells). The missing y-axis title in Figure 6C has been added, and the Figure 6 legend has been revised to better clarify the correspondence between Figure 6A and Figures 6B/C.
Reviewer Comment 6: The conclusion of the result section 3.6 “These data indicate that talazoparib increases the latency reversal effect of vorinostat in HIV-1 latently infected human CD4 T-cells and subsequently eliminates reactivated cell populations through the activation of NK cell activity.” is only partially sustained by data. Authors have shown an increased NK activity upon treatment of PBMCs with talazoparib and vorinostat, against K562 cells, not against reactivated cell population.
Authors Response 6: According to reviewer’s suggestion, we have toned down this conclusion, and the revised sentence has been added in lines 403–406 on page 13-14.

Reviewer 2 Report
Comments and Suggestions for Authors
The study by Tibeb et al. investigates the potential to enhance the reactivation of the latent HIV reservoir through the combined use of HDAC inhibitors and PARP inhibitors, aiming to strengthen the "kick and kill" therapeutic strategy. The study is well-presented, and the results open new possibilities for optimizing this therapeutic approach.
Comments
Materials and Methods. Table 1: Why is vorinostat, an HDAC inhibitor, listed among the β-catenin inhibitors?
Results. Section 3.5: In PBMCs, the combination of an HDAC inhibitor and a PARP inhibitor leads to a reduction in latently infected cells but, paradoxically, also a decrease in activated cells. A possible explanation is the activation of NK cells, which may induce the elimination of activated cells. It would be important for the authors to include cell viability data for PBMC cultures treated with the individual inhibitors and their combination, using also flow cytometry analysis before and after treatment to determine which cell fraction is depleted.
Discussion: Well-structured, though partially exceeding the study's direct scope. A more concise focus on the results and their significance is recommended.
Comments on the Quality of English Language
The English is fine
Author Response
We sincerely appreciate your time in reviewing our manuscript and providing valuable feedback. Below is our response to each comment.
Reviewer Comment 1: Materials and Methods. Table 1: Why is vorinostat, an HDAC inhibitor, listed among the β-catenin inhibitors?
Authors Response 1: Thank you for pointing out the mistake in Table 1. We have corrected it accordingly.
Reviewer Comment 2: Results. Section 3.5: In PBMCs, the combination of an HDAC inhibitor and a PARP inhibitor leads to a reduction in latently infected cells but, paradoxically, also a decrease in activated cells. A possible explanation is the activation of NK cells, which may induce the elimination of activated cells. It would be important for the authors to include cell viability data for PBMC cultures treated with the individual inhibitors and their combination, using also flow cytometry analysis before and after treatment to determine which cell fraction is depleted.
Authors Response 2: In flow cytometry, dead PBMCs typically exhibit altered light scatter properties compared to viable cells. As a result, dead cells tend to cluster in regions of the FSC vs. SSC plot associated with debris, often near the lower left corner. Therefore, our lymphocyte gating may not capture all dead cell populations. To address this, we also extracted all live cell populations using a Live/Dead staining dye to exclude early dead cells that still retain cell morphology. Therefore, when the number of live infected cells decreases due to drug treatment, it may indicate an increase in the number of dead cells. To clarify this, we have also added our gating strategy for lymphocytes, single cells, and Live/Dead cell populations before categorizing CD4 T cells in Figure 6A. The figure legend is also revised accordingly.
Reviewer Comment 3: Discussion: Well-structured, though partially exceeding the study's direct scope. A more concise focus on the results and their significance is recommended.
Authors Response 3: Thank you for your positive comments. We believe that a detailed discussion of both Hippo signaling and Wnt/β-catenin signaling in the context of HIV-1 latency reversion and subsequent elimination is essential, as both pathways are inhibited by PARP inhibitors. Therefore, we would like to retain our original sentences in this section, especially since the reviewer acknowledged the well-structured discussion in the manuscript.

Reviewer 3 Report
Comments and Suggestions for Authors
Authors present clear and precise elegant in vitro studies to define the synergistic potential of the combination of HDAC inhibitors and PARP inhibitors to advance HIV Cure strategy.
Results warrant further investigation to determine in vivo safety and efficacy.
Importantly, e FDA approval of both HDAC 526
and PARP inhibitors for cancer treatment, this pre-existing approval streamlines the path- 527
way for clinical trials in the context of HIV cure strategies
Author Response
We sincerely appreciate your insightful review of our manuscript and your encouraging comments.
Reviewer Comment: Authors present clear and precise elegant in vitro studies to define the synergistic potential of the combination of HDAC inhibitors and PARP inhibitors to advance HIV Cure strategy.
Results warrant further investigation to determine in vivo safety and efficacy.
Importantly, e FDA approval of both HDAC 526
and PARP inhibitors for cancer treatment, this pre-existing approval streamlines the path- 527
way for clinical trials in the context of HIV cure strategies
Authors Response: We appreciate your positive comment. We are highly encouraged to further investigate the efficacy of this combination therapy in an in vivo study. Attached is the revised manuscript that includes all of the invaluable suggestions and comments from the reviewers.

Round 2
Reviewer 1 Report
Comments and Suggestions for Authors
Authors have succesfully addressed all raised issues.